# Virtual care use during the COVID-19 pandemic and its impact on healthcare utilization in patients with chronic disease: A population-based repeated cross-sectional study

**Vess Stamenova**[1]*, **Cherry Chu**[1], **Andrea Pang**[2], **Jiming Fang**[2], **Ahmad Shakeri**[1], **Peter Cram**[2,3], **Onil Bhattacharyya**[1,4], **R. Sacha Bhatia**[1,2], **Mina Tadrous**[1,5]

**1** Women's College Hospital Institute for Health Systems Solutions and Virtual Care, Women's College Hospital, Toronto, Ontario, Canada, **2** ICES, Toronto, Ontario, Canada, **3** University of Texas Medical Branch at Galveston, Galveston, Texas, United States of America, **4** Department of Family & Community Medicine, University of Toronto, Toronto, Ontario, Canada, **5** Leslie Dan Faculty of Pharmacy, University of Toronto, Toronto, Ontario, Canada

* vess.stamenova@wchospital.ca

## Abstract

### Purpose

It is currently unclear how the shift towards virtual care during the 2019 novel coronavirus (COVID-19) pandemic may have impacted chronic disease management at a population level. The goals of our study were to provide a description of the levels of use of virtual care services relative to in-person care in patients with chronic disease across Ontario, Canada and to describe levels of healthcare utilization in low versus high virtual care users.

### Methods

We used linked health administrative data to conduct a population-based, repeated cross-sectional study of all ambulatory patient visits in Ontario, Canada (January 1, 2018 to January 16, 2021). Further stratifications were also completed to examine patients with COPD, heart failure, asthma, hypertension, diabetes, mental illness, and angina. Patients were classified as low (max 1 virtual care visit) vs. high virtual care users. A time-series analysis was done using interventional autoregressive integrated moving average (ARIMA) modelling on weekly hospitalizations, outpatient visits, and diagnostic tests.

### Results

The use of virtual care increased across all chronic disease patient populations. Virtual care constituted at least half of the total care in all conditions. Both low and high virtual care user groups experienced a statistically significant reduction in hospitalizations and laboratory testing at the start of the pandemic. Hospitalization volumes increased again only among

**Data Availability Statement:** The data is not publicly available and access is limited to the Institute for Clinical Evaluative Sciences (ICES)

(https://www.ices.on.ca/), which is a prescribed entity under section 45 of Ontario's Personal Health Information Protection Act (PHIPA). Researchers, students, policy makers or knowledge users who are affiliated with a publicly funded, not-for-profit organization and who want to obtain and analyze ICES data to answer a research question may submit a request to ICES DAS (https://www.ices.on.ca/DAS/Public-Sector). DAS staff will contact the requestor to discuss the project's feasibility, timeline and cost. Projects requesting access to data require the approval of a research ethics board. We are happy to share our data creation plan specifying our analyses if you contact the corresponding author and in turn you can use the information in the data creation plan to run the same or follow-up analyses. A list of all datasets used is available in the paper. All billing codes used are listed in the Supporting information files.

**Funding:** This work was funded through the Ontario Ministry of Health as funds provided to the Centre for Digital Health Evaluation at Women's College Hospital, Toronto, Ontario. The funders had no role in study design, data collection and analysis, decision to publish, or preparation of the manuscript.

**Competing interests:** The authors have declared that no competing interests exist.

the high users, while testing increased in both groups. Outpatient visits among high users remained unaffected by the pandemic but dropped in low users.

## Conclusion

The decrease of in-person care during the pandemic was accompanied by an increase in virtual care, which ultimately allowed patients with chronic disease to return to the same visit rate as they had before the onset of the pandemic. Virtual care was adopted across various chronic conditions, but the relative adoption of virtual care varied by condition with highest rates seen in mental health.

## Introduction

The 2019 novel coronavirus (COVID-19) pandemic has forced healthcare systems to balance the risk of COVID infection with the potential negative impacts of delaying care [1, 2]. This has led to a significant adoption of virtual care services globally as a means to continue seeing patients while minimizing the cost of contact [3–5]. In Ontario, Canada, in the first 3 months of the pandemic 70% of all ambulatory visits were conducted virtually (telephone or video) [3, 6] and 86% of physicians conducted at least one virtual care visit [3].

With unprecedented levels of virtual care use, policy makers and payors are concerned about the quality of care being delivered virtually and potential increases in costs due to higher healthcare utilization. The COVID-19 pandemic has also led to challenges in the care of patients with chronic disease [7, 8]. A study from the USA reported a 90% reduction in rates of screening and prevention services [7]. Similarly, in Ontario, Canada, a study reported 89% fewer preventative primary care visits [9]. Additionally, reports of hospitalizations across various jurisdictions have demonstrated that there was a general decline in hospitalizations across many chronic conditions, especially early in the pandemic [10–12]. Such reductions in in-person visits and hospitalizations may lead to higher rates of future healthcare utilization during later stages of the pandemic (e.g. hospitalizations, emergency department visits).

It is currently unclear how the shift towards virtual care during the pandemic may have impacted chronic disease management and if virtual care was able to compensate for pandemic-related drops in in-person visits. The extent at which virtual care adoption was maintained throughout later stages of the pandemic is also still unknown. In Ontario, the most populated province in Canada, the healthcare system is publicly funded and the data we used covers all healthcare services used [13]. It, therefore, presents an opportunity to examine shifts in care at a population-level.

The goals of our study were to provide a description of the levels of use of virtual care services, relative to in-person care in patients with chronic disease across Ontario, Canada and to describe levels of healthcare utilization in low versus high virtual care users.

## Methods

This study received an ethics exemption provided by the Research Ethics Board of Women's College Hospital (REB # 2020-0106-E). This study will be conducted at the Institute for Clinical Evaluative Sciences (ICES), which is a prescribed entity under section 45 of Ontario's Personal Health Information Protection Act (PHIPA). The study will use only data that is already de-identified and will be conducted under ICES' stringent privacy regulations. It has been

determined that studies/projects that fall under Section 451 of the Personal Health Information Protection Act: Disclosure for planning and management of health system do not require REB review and approval.

## Study design

We used linked health administrative data to conduct a population-based, repeated cross-sectional study of all ambulatory patient visits in Ontario, Canada beginning January 1, 2018 and extending to the 2nd week of January, 2021. We identified visits weekly in the total Ontario population, and for subpopulations of patients with chronic disease.

**Data sources.** We used the following administrative databases: the Ontario Health Insurance Plan (OHIP) for physician claims, the Canadian Institutes of Health Information Discharge Abstract Database (CIHI-DAD) for information about all hospitalizations, the CIHI National Ambulatory Care Reporting System (NACRS) for hospital- and community-based ambulatory care including ED visits, the Ontario Drug Benefit Database (ODB) for prescription medication data for those ≥65 years old, the ICES Physician Database (IPDB) for data on physician specialty, the Ontario Mental Health Reporting Systems (OMHRS) for psychiatric hospitalizations and the Ontario Laboratories Information System (OLIS) for community- and hospital-based laboratories. Databases were linked using unique identifiers and analyzed at ICES (the Institute for Clinical Evaluative Sciences). Together these databases cover all of healthcare activity in Ontario. Use of these databases for the purposes of this study was authorized under §45 of Ontario's Personal Health Information Protection Act, which does not require review by a research ethics board (REB). Nonetheless, the study also received an REB exemption approval from Women's College Hospital REB (REB # 2020-0106-E). All data was de-identified at ICES and individual patient consent was waived.

## Population

For each week of our study period (Jan 1, 2018- Jan 16, 2021), we identified all ambulatory care visits (in-person and virtual care visits) over the study period using relevant provider billing codes (see S1 File for full list of billing codes). We excluded claims for any patient who was a non-Ontario resident and/or had an invalid or missing health card number.

Further stratifications were also completed for circumscribed patient populations: COPD, heart failure, asthma, hypertension, and diabetes if they had at least one entry in the corresponding disease-specific registry at ICES [13] any time prior to their index visit (their first visit during the observation window). Patients with mental illness were identified by at least one outpatient claim with a primary care provider visit linked to psychiatric diagnostic codes, or a mental health service code or any code by a psychiatrist in the past 3 years. Angina patients were identified by at least one ED visit with the relevant ICD-9 or 10 code in the past 12 months (see S1 File for details on diagnostic codes used in the definitions).

## Analysis

For each week of our study period (Jan 1, 2018- Jan 16, 2021) we examined the number and rate per 1000 of in-person and virtual care visits across all age groups and the percentage of ambulatory visits that were virtual (versus in-person) for all patients eligible for healthcare services in Ontario.

To examine differences in healthcare utilization between patients who use virtual care and those who do not, we classified patients into two groups based on their use of virtual care. Low virtual care users had at least one visit (virtual or in-person) after the onset of the pandemic (March 14, 2020) and they could have a maximum of one virtual care visit during the entire

period. Patients who had no virtual care visits were also included in the low virtual care group, but they had to have at least one in-person visit in order to be included. This means we excluded patients who did not receive care at all during the pandemic period (after March 14, 2020). Patients in the high virtual care use group had to have at least 2 virtual care visits. There was no limit on the number of in-person visits patients could have in either group. To determine the level of healthcare utilization in each group, we examined the number of weekly hospitalizations, total ambulatory (outpatient) visits, ED visits, diagnostic tests, lab tests, and prescriptions for those over 65 years of age, from January 2018 to September 2020. A time-series analysis was done using interventional autoregressive integrated moving average (ARIMA) modelling. Interventional ARIMA is a common technique used to better understand the impact of a significant event to time series data and to forecast future observations while accounting for components such as trend, seasonality, and autocorrelation [14]. A step function beginning on March 14, 2020 was applied to the model in order to determine whether there were significant shifts in utilization in each of the two groups before versus during the pandemic.

## Results

### Overall virtual care use

Between January 1, 2018 and January 15, 2021 there were just over 215 million virtual and in-person ambulatory care visits (73 460 386 in 2018, 73 629 600 in 2019 and 68 032 404 in 2020). Of these, 33 million visits were conducted virtually (933 099 in 2018, 1 234 082 in 2019 and 30 858 723 in 2020). The majority of virtual visits across the entire period (93%) occurred during the pandemic after the temporary virtual codes were introduced by the Ontario Ministry of Health in March 2020. Between March 15, 2020 and January 15, 2021, there were just over 32 million virtual visits completed. The average rate of virtual care visits was 48 visits per 1000 patients per week. The average rate of in-person visits was 34 visits per 1000 per week. In comparison, the average rate of virtual care before the pandemic was 1.4 visits per 1000 patients per week.

Between March 15 and July 1, 2021, total ambulatory care reduced by 22%, in-person visits reduced by 75%, and 69% of ambulatory care occurred virtually (12.1 million visits). Virtual care never dropped below 50% of all ambulatory care throughout the pandemic period (Fig 1).

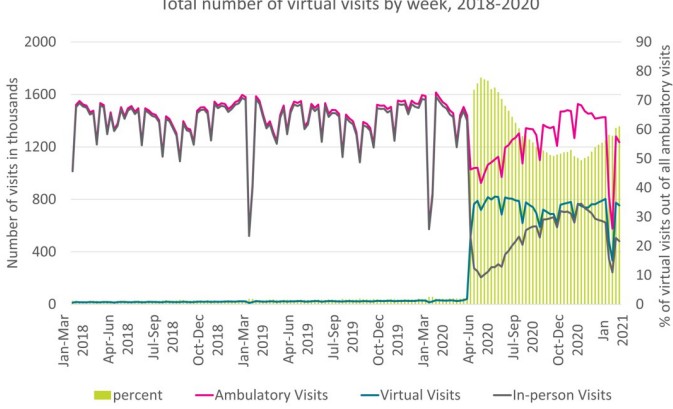

**Fig 1. Total number of virtual and in-person visits (line) and percent virtual care use out of total ambulatory care (bars).** *All ambulatory visits = both in-person and virtual.

The majority of virtual care (94%) was delivered through the new billing codes, which allowed for phone or video visits through any platform.

On a weekly basis, an average of 20 006 providers out of 23 835 practicing physicians (84%) provided at least one virtual care visit. Psychiatrists had the highest rates of virtual care with about 90% of care being virtual in the early months of the pandemic and mostly staying above 80% throughout the entire pandemic.

## Virtual care use among patients with chronic disease

The use of virtual care increased in March 2020 across all chronic disease patient populations examined (Fig 2). The highest rates of virtual care visits were seen in CHF (151 visits per week per 1000 patients), COPD (126 visits per week per 1000 patients), and angina patients (116 visits per week per 1000 patients) (Table 1). The increase in virtual care use was associated with a sharp decline of in-person visits in March due to the start of the first Ontario lockdown. Across all conditions examined, in-person care decreased by 72–93%. This sharp decrease was followed by a gradual increase in in-person visits, but on average for the entire period examined after March 14, virtual care constituted at least half of the total care (55% of total care in CHF, 56% in COPD, 58% in hypertension and diabetes, 62% in asthma and angina and 72% in mental health care).

## Healthcare utilization among low and high virtual care users

**Hospitalizations.** High virtual care users had more hospitalizations than low virtual care users (average weekly volumes across conditions ranged from 1306–4205 admissions in high users versus 196–817 admissions in low users). Trends were similar across the four patient populations. Both low and high virtual care user groups experienced a statistically significant reduction in hospitalizations at the start of the pandemic on Mar 14, 2020 (from one month before to one month after Mar 14, average weekly volumes across conditions dropped from 233–896 to 176–605 admissions in low users and from 1496–5539 to 1040–3791 admissions in high users, p < .0001) (Fig 3). Despite the significant drop at the beginning of the pandemic, hospitalization volumes began to increase again among the high users (1379–5245 visits in first week of June 2020) as the pandemic progressed but hospitalization remained low among the low users (167–678 admissions in first week of June 2020). The percent change in the average number of hospitalizations from pre to post Mar 14, 2020 is shown in Table 2.

**Outpatient visits.** High virtual care users had higher outpatient visit volumes than low users (average weekly volumes across conditions ranged from 41,164–382,929 visits in high users versus 5865–68,724 visits in low users). Across all four patient populations, there was a statistically significant drop in visits among low virtual care users at the onset of the pandemic (from one month before to one month after Mar 14, average weekly volumes across conditions dropped from 6484–77,274 to 3794–44,062 visits, p < .0001), and visit volumes remained stagnant afterwards (4213–46,909 visits in first week of June 2020) (Fig 4). However, the number of outpatient visits among high users remained unaffected by the pandemic (44,777–421,875 visits at one month before to 42,407–402,199 visits at one month after Mar 14, p>0.5). High volumes were maintained throughout, and even appeared to be increasing during the pandemic (51,656–489,404 in first week of June 2020).

**Laboratory testing.** Laboratory claims were higher overall among high virtual care users than low users (average weekly volumes across conditions ranged from 83,480–606,752 tests in high users versus 14,533–128,354 tests in low users). Both high user and low user patients across all four conditions experienced a significant drop in testing volumes coinciding with the start of the pandemic (from one month before to one month after Mar 14, average weekly

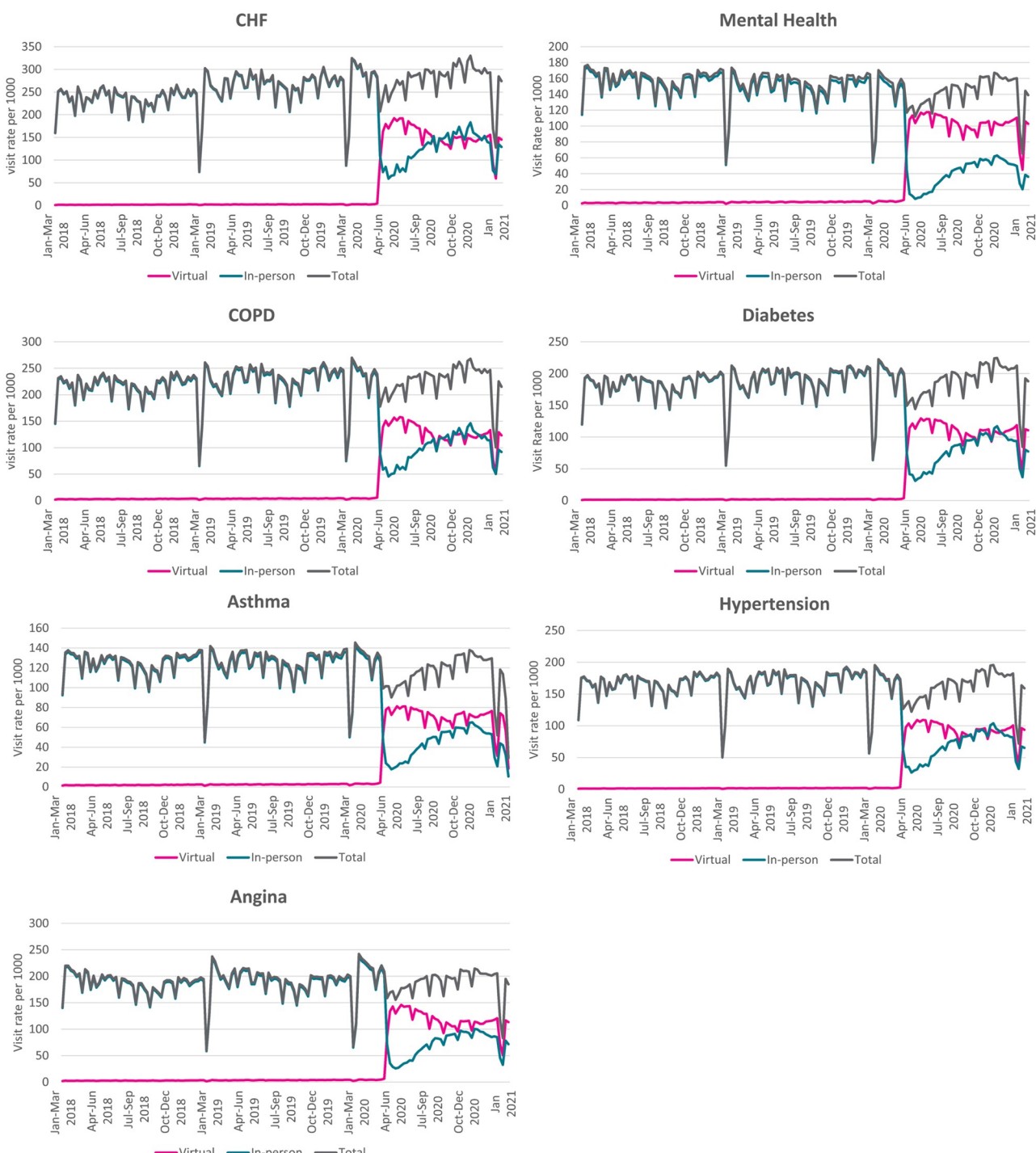

**Fig 2. Weekly rate of virtual and in-person care visits per 1000 by medical condition.**

volumes across conditions dropped from 16,230–155,905 to 5462–38,297 tests in low users and from 95,972–756,319 to 39,506–264,035 tests in high users, p < .0001) (Fig 5). However, volumes appeared to gradually increase again in both user groups as the pandemic progressed (in the first week of June 2020, low users: 9898–75,890 tests and high users: 82,270–604,512 tests).

**Table 1. Average weekly visit rates per 1000 patients by modality (virtual or in-person) before (Jan 2018 to Mar 2020) and during (Mar 2020 to Jan 2021) the pandemic across chronic conditions.**

| | COPD | | | CHF | | | Mental Health | | |
|---|---|---|---|---|---|---|---|---|---|
| | Before COVID-19 | During COVID-19 | % change | Before COVID-19 | During COVID-19 | % change | Before COVID-19 | During COVID-19 | % change |
| **Virtual** | 3.4 | 126.2 | 3612 | 2.0 | 151.3 | 7465 | 4.0 | 101.6 | 2440 |
| **In-Person** | 220.7 | 98.5 | -55.4 | 250.6 | 124.6 | -50.3 | 150.3 | 39.8 | -73.5 |
| **Total** | 224.1 | 224.7 | 0.3 | 252.6 | 275.9 | 9.2 | 154.3 | 141.4 | -8.4 |
| | Angina | | | Diabetes | | | | | |
| | Before COVID-19 | During COVID-19 | % change | Before COVID-19 | During COVID-19 | % change | | | |
| **Virtual** | 3.4 | 116.4 | 3324 | 1.8 | 108.8 | 5944 | | | |
| **In-Person** | 187.3 | 70.3 | -62.5 | 184.6 | 77.6 | -58.0 | | | |
| **Total** | 190.6 | 186.7 | -2.0 | 186.4 | 186.4 | 0 | | | |
| | Asthma | | | Hypertension | | | | | |
| | Before COVID-19 | During COVID-19 | % change | Before COVID-19 | During COVID-19 | % change | | | |
| **Virtual** | 2.3 | 70.7 | 2974 | 1.6 | 92.2 | 5663 | | | |
| **In-Person** | 122.9 | 44.1 | -64.1 | 165.5 | 67.9 | -59.0 | | | |
| **Total** | 125.2 | 114.8 | -8.3 | 167.1 | 160.0 | -4.2 | | | |

The p,d,q values of the best fit ARIMA models are reported in S1 File. Figures showing emergency department visits, prescriptions for those over 65 years of age and total patient cost for low and high virtual care users are also provided in S1 File.

## Discussion

Our study examining data over the first 9 months of the COVID-19 pandemic showed that virtual care adoption (telephone and video) constituted at least 50% of all ambulatory care in Ontario, Canada. The strong adoption of virtual care in combination with some return of in-person care over later stages of the pandemic allowed physicians to maintain adequate levels of care for chronic disease patients during the pandemic. Virtual care was adopted across all chronic conditions examined, but the degree of uptake of virtual care varied. Differences in adoption rates of virtual care need to be considered when designing best practices of care and policies surrounding the continued use of virtual care. Patients who used more virtual care appeared to be higher users of the healthcare system in general before and during the pandemic. This suggests that higher use patients who had access to the healthcare system before the pandemic received similar access to the healthcare system during the pandemic though a combination of care services including virtual care. The percentage of visits that were virtual in Ontario, Canada (59%) was higher than rates reported in other countries such as the USA (30%) and Australia (42%) [4, 15, 16]. While there was some increase of in-person care in the May to September of 2020, the high adoption of virtual care was largely maintained throughout waves of the pandemic. Psychiatry had the highest rates of virtual care use during the pandemic. The use of any virtual care in psychiatry is enabled to a greater extent than other specialties simply due to the fact that most psychiatry visits do not require a physical examination [17]. Interestingly, while psychiatry previously reported 3 times the average rates of virtual care use in Ontario before the pandemic, these rates reduced to about double the rate of virtual

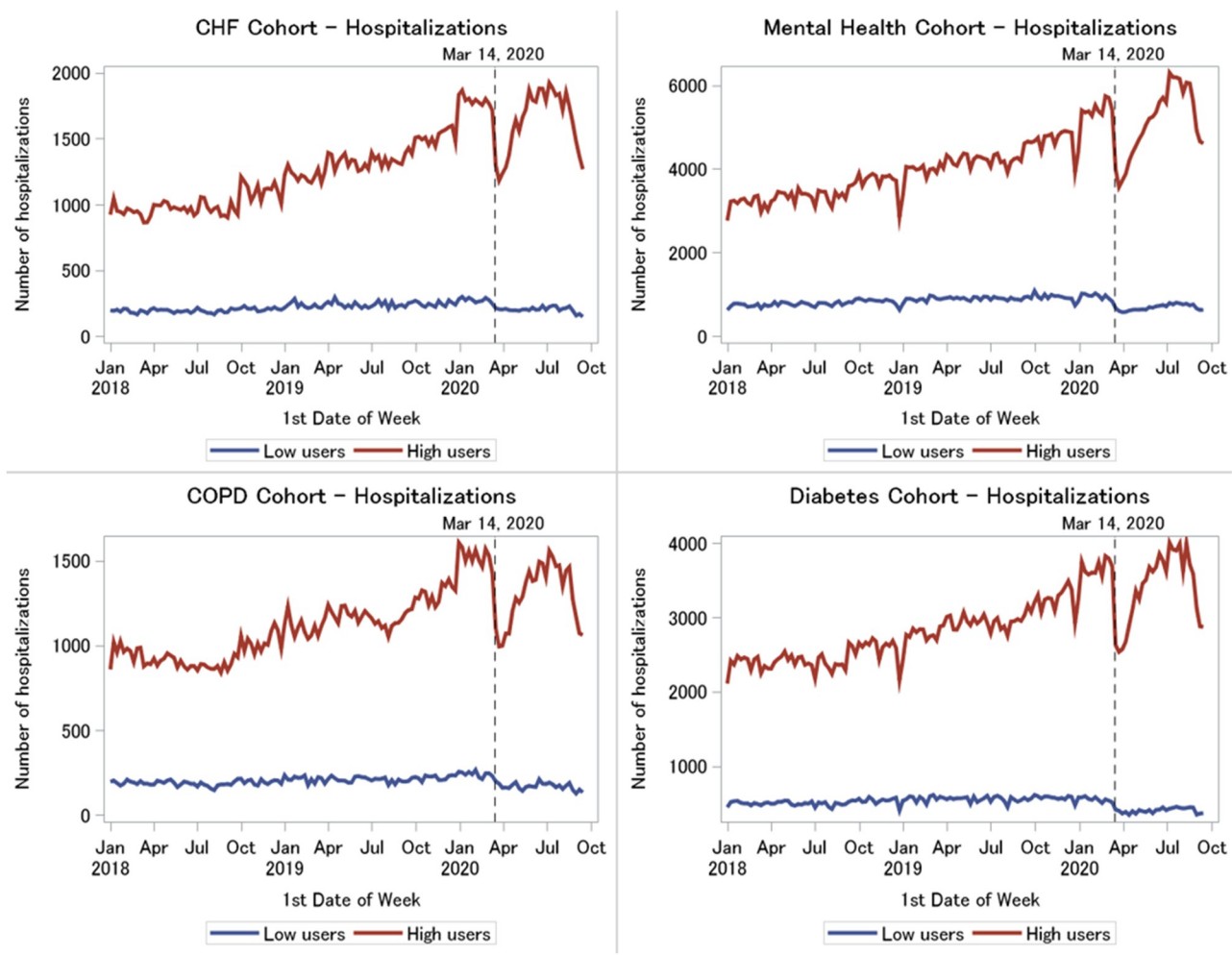

**Fig 3. Weekly hospitalizations in high versus low virtual care users by medical condition, January 2018 to September 2020.**

care use during the pandemic. This relative reduction is likely due to the substantial shift to virtual care across other patient populations who previously did not have access to virtual care.

The pandemic led to an overall decrease of total ambulatory care (virtual or in-person) across all chronic diseases studied. This reduction in ambulatory visits and testing has been reported elsewhere [7, 10], with some reporting more severe impacts in diabetes management

**Table 2. Percent change in the average number of hospitalizations, outpatient visits and lab testing from pre to post Mar 14, 2020.**

|  | Hospitalizations | | Outpatient Visits | | Lab Testing | |
|---|---|---|---|---|---|---|
|  | Low Users | High Users | Low Users | High Users | Low Users | High Users |
| **CHF** | -9.4% | 32.2% | -31.5% | 27.5% | -34.3% | -5.6% |
| **MH** | -19.4% | 28.1% | -34.0% | 31.0% | -42.4% | -6.9% |
| **COPD** | -17.2% | 17.3% | -38.7% | 14.5% | -42.2% | -17.1% |
| **Diabetes** | -24.8% | 21.4% | -39.1% | 15.1% | -39.6% | -16.9% |

(Low users = minimum of 1 ambulatory in-person or virtual visit but maximum of 1 virtual visit after March 14, 2020; High users = minimum of 2 virtual visits after March 14, 2020).

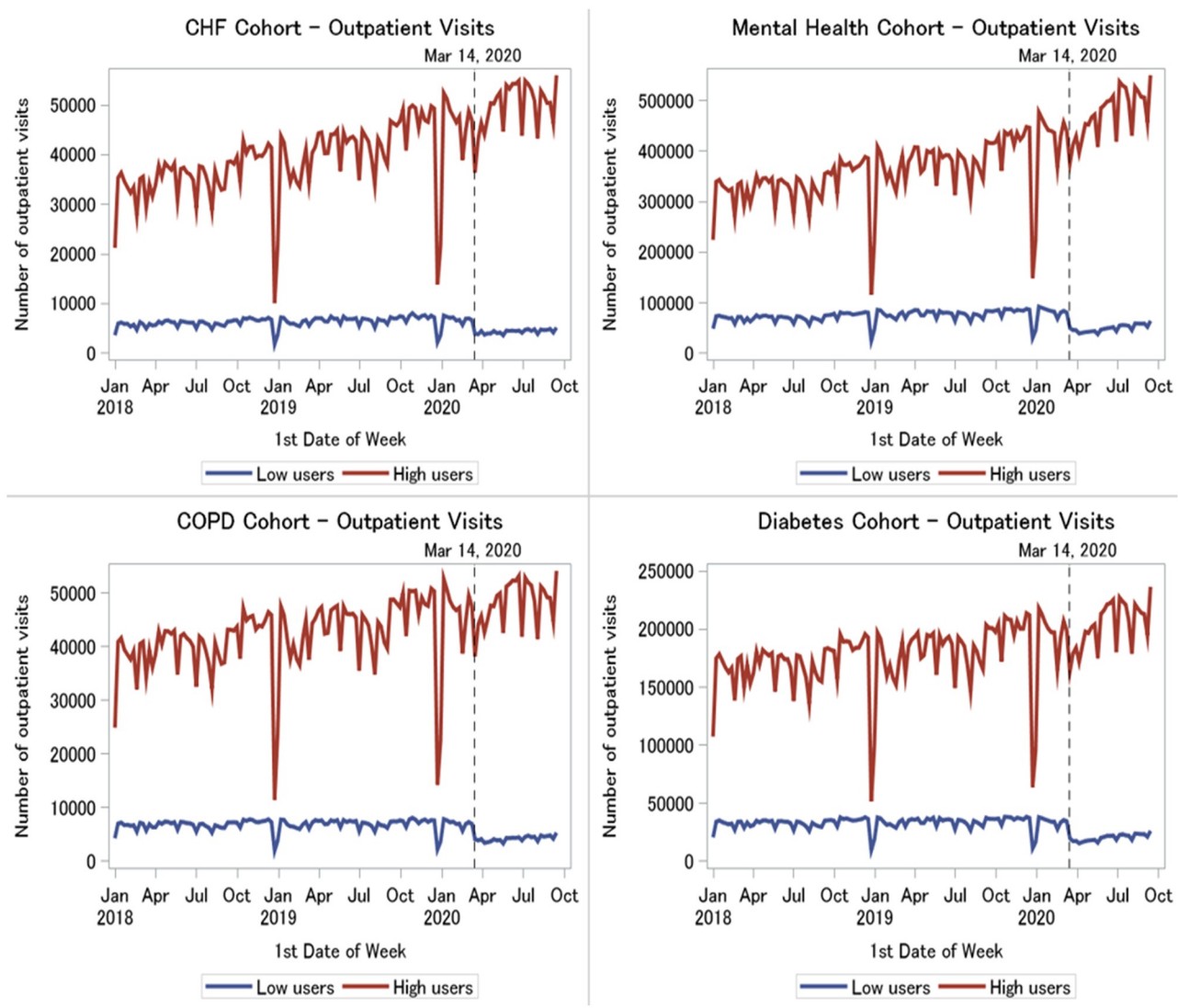

**Fig 4. Weekly outpatient visits in high versus low virtual care users by medical condition, January 2018 to September 2020.**

[8]. It has been suggested that the causes of these reductions in utilization are multifactorial and include patient avoidance of care, increased threshold of hospitalizations from providers, and changes in lifestyle and self-management in the context of lockdown measures and social distancing [10]. The decrease in total ambulatory care was a result of a very sharp drop of in-person care due to COVID-19 restrictions. Restrictions on in-person care, however, were accompanied by an increase in virtual care, which ultimately allowed patients with chronic disease to return to the same visit rate as they had before the onset of the pandemic and this finding is consistent with reports from other jurisdictions [18, 19]. This was likely due to both patients and providers adopting virtual care as part of their routine care and enabling the conversion of pre-pandemic in-person visits into virtual visits where possible [19].

The early weeks of the pandemic were also accompanied by reductions in hospitalization across both high and low virtual care users, while outpatient care reduced only for low virtual care users. Given both acute and outpatient care use was higher before the pandemic for high virtual care users relative to that of low virtual care users, it is likely that these patients were

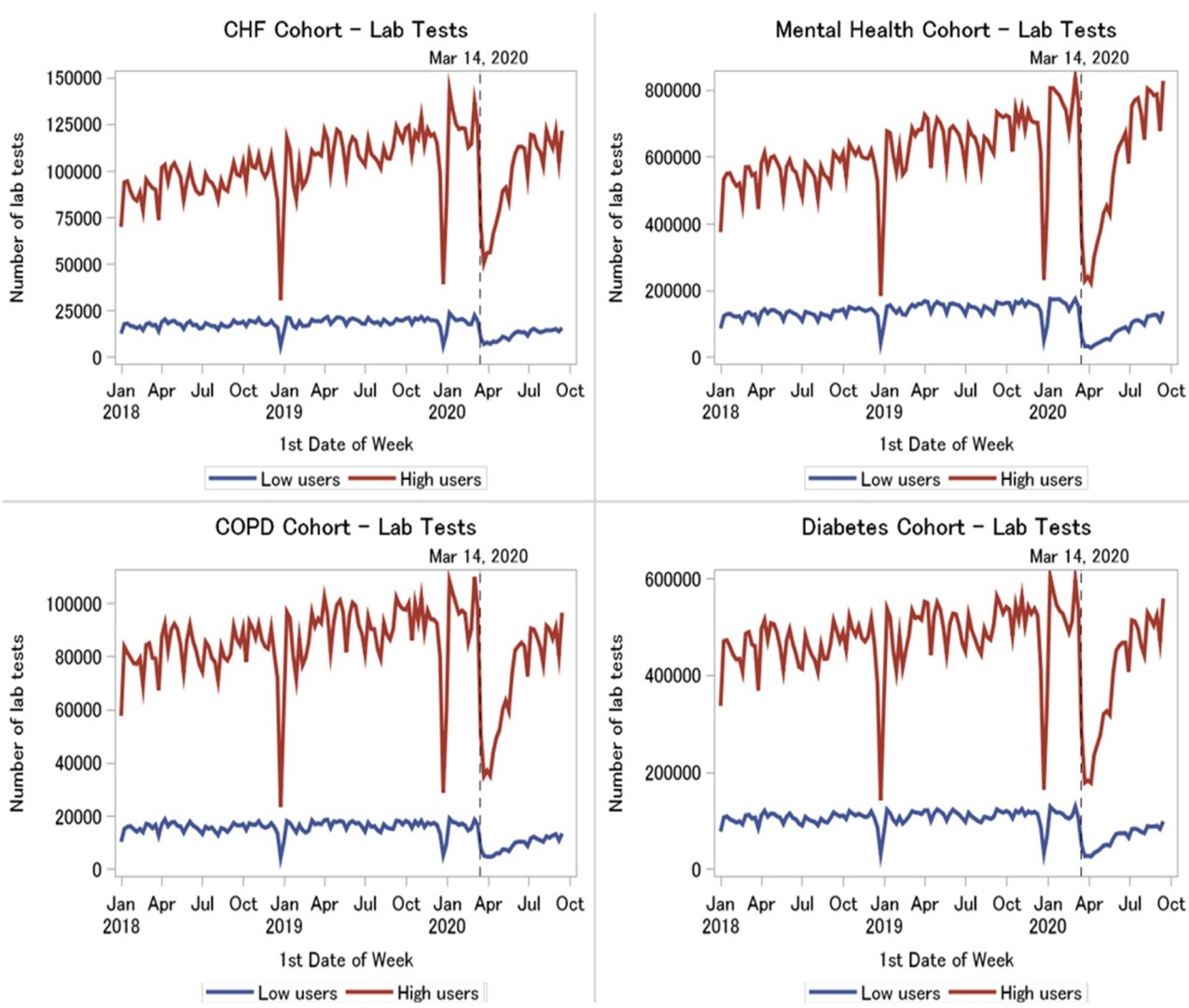

**Fig 5. Weekly laboratory testing in high versus low virtual care users by medical condition, January 2018 to September 2020.**

sicker before the pandemic and required continuous care during the pandemic. As more than half of the outpatient care was delivered virtually, the outpatient care that these patients received would have included a substantial virtual component. This finding supports the idea that virtual care afforded continued access to care for patients who were higher users of the healthcare system and likely sicker. This finding is consistent with reports that patients with greater disease burden having greater use of virtual care services during the pandemic [5, 18]. More stable patients with chronic disease had some contact with the system but were likely advised to hold off on care until the risk of infection from COVID-19 reduces. This reduction on non-urgent care was likely partially due to restrictions on elective procedures in the province in effect during the early stages of the pandemic, but it may also be a result of changes in physician prescribing and testing practices [18].

## Limitations

Study limitations include limitations associated with health administrative data analysis, such as reliance on diagnostic billing codes and a lack of clinical details leading to an inability to

report on what the reasons for visits were. Most chronic disease diagnostic codes have been validated [13], which improves our confidence. The biggest limitation of this study is that the data is collected in the context of a pandemic. It is unclear to what extent some of the relationships we see with virtual care are due to the fact that patients were receiving virtual care or due to the pandemic itself, where virtual care was likely the predominant method to receive any care. These findings should be re-examined once the pandemic is over and both in-person and virtual care are equally accessible. Finally, as the billing codes in Ontario do not distinguish between video and phone, we were unable to report with confidence on relative use of telephone versus video.

## Conclusions

In conclusion, in this population-based, repeated cross-sectional study in the largest province of Canada, where a universal healthcare system supports reimbursements of virtual care during the pandemic, we find that virtual care was adopted across various chronic conditions, but the relative adoption of virtual care varied by condition. Further, more than half of ambulatory care was virtual. allowing for regular visits for patients with chronic conditions to be maintained during the pandemic. Patients who became greater users of virtual care were generally high users of the healthcare system before and during the pandemic, suggesting that virtual care provided continued access to care for patients who were higher users and potentially sicker. These findings suggest that proper frequency of visits of chronic disease patients can be maintained through a mix of in-person and virtual visits even in cases where the disease severity is higher. Long-term studies should examine, however, whether the quality of care received through a mixed in-person and virtual care model is the same as that received through in-person care alone in order to inform policy decisions about their continued use in the healthcare system.

## Supporting information

**S1 File.**
(DOCX)

## Author Contributions

**Conceptualization:** Vess Stamenova, Cherry Chu, Ahmad Shakeri, R. Sacha Bhatia.

**Data curation:** Andrea Pang, Jiming Fang.

**Formal analysis:** Cherry Chu, Andrea Pang, Jiming Fang.

**Funding acquisition:** Vess Stamenova, R. Sacha Bhatia, Mina Tadrous.

**Investigation:** R. Sacha Bhatia, Mina Tadrous.

**Methodology:** Vess Stamenova, Cherry Chu, Ahmad Shakeri, Peter Cram, Onil Bhattacharyya, R. Sacha Bhatia, Mina Tadrous.

**Project administration:** Vess Stamenova.

**Supervision:** R. Sacha Bhatia, Mina Tadrous.

**Validation:** Vess Stamenova, Mina Tadrous.

**Visualization:** Vess Stamenova, Mina Tadrous.

**Writing – original draft:** Vess Stamenova, Cherry Chu.

**Writing – review & editing:** Vess Stamenova, Cherry Chu, Andrea Pang, Jiming Fang, Ahmad Shakeri, Peter Cram, Onil Bhattacharyya, R. Sacha Bhatia, Mina Tadrous.

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
