## [Decision Letter · Decision Letter 0]

2 Dec 2021

PONE-D-21-24910Virtual care use during the COVID-19 pandemic and its impact on healthcare utilization in patients with chronic disease: a population-based repeated cross-sectional study.PLOS ONE

Dear Dr. Stamenova,

Thank you for submitting your manuscript to PLOS ONE. After careful consideration, we feel that it has merit but does not fully meet PLOS ONE’s publication criteria as it currently stands. Therefore, we invite you to submit a revised version of the manuscript that addresses the points raised during the review process.Please, address the points raised by the reviewers. However, take into account the PLOS ONE criteria for publication. To be accepted in PLOS ONE, manuscripts must be technically sound and the work must be conducted with analytical rigor, but the PLOS ONE criteria do not include “sufficiently novel”.Revise the references to include other scholars studies related to the use of virtual clinic during COVID-19.Besides, I have other minor points:The work of Dr Bhatia (ref 1) discusses about the potential risks of deferring medical care, but not specifically about ACSC.Line 127: the sum of the four groups of visits is 215 million, instead of 218There is a typo in the numbering of tables (table 2 appears before table 1)Please submit your revised manuscript by Jan 16 2022 11:59PM. If you will need more time than this to complete your revisions, please reply to this message or contact the journal office at plosone@plos.org. Please include the following items when submitting your revised manuscript:A rebuttal letter that responds to each point raised by the academic editor and reviewer(s). You should upload this letter as a separate file labeled 'Response to Reviewers'.A marked-up copy of your manuscript that highlights changes made to the original version. You should upload this as a separate file labeled 'Revised Manuscript with Track Changes'.An unmarked version of your revised paper without tracked changes. You should upload this as a separate file labeled 'Manuscript'.If applicable, we recommend that you deposit your laboratory protocols in protocols.io to enhance the reproducibility of your results. Protocols.io assigns your protocol its own identifier (DOI) so that it can be cited independently in the future. For instructions see: https://journals.plos.org/plosone/s/submission-guidelines#loc-laboratory-protocols. Additionally, PLOS ONE offers an option for publishing peer-reviewed Lab Protocol articles, which describe protocols hosted on protocols.io. Read more information on sharing protocols at https://plos.org/protocols?utm_medium=editorial-email&utm_source=authorletters&utm_campaign=protocols.

We look forward to receiving your revised manuscript.

Kind regards,

Juan F. Orueta, MD, PhD

Academic Editor

PLOS ONE

https://journals.plos.org/plosone/s/file?id=ba62/PLOSOne_formatting_sample_title_authors_affiliations.pdf"

“This work was funded through the Ontario Ministry of Health as funds provided to the Centre for Digital Health Evaluation at Women's College Hospital, Toronto, Ontario.”

Reviewers' comments:

Reviewer's Responses to Questions

**Comments to the Author**

1. Is the manuscript technically sound, and do the data support the conclusions?

Reviewer #1: Yes

Reviewer #2: Yes

Reviewer #3: No

Reviewer #4: Yes

2. Has the statistical analysis been performed appropriately and rigorously? 

Reviewer #1: I Don't Know

Reviewer #2: Yes

Reviewer #3: No

Reviewer #4: Yes

3. Have the authors made all data underlying the findings in their manuscript fully available?

Reviewer #1: Yes

Reviewer #2: No

Reviewer #3: No

Reviewer #4: No

4. Is the manuscript presented in an intelligible fashion and written in standard English?

Reviewer #1: Yes

Reviewer #2: Yes

Reviewer #3: Yes

Reviewer #4: Yes

5. Review Comments to the Author

Reviewer #1: General Comments

Thank you for the opportunity to review this paper. Overall, I thank the authors for a well-written manuscript on charting the use of virtual healthcare services across the most common chronic diseases in Ontario both pre- and post-pandemic. As the COVID-19 situation continues, and the world adapts gradually to a post-pandemic new normal, I agree with the authors that it is timely to understand rates of virtual healthcare utilisation stratified by the various chronic care conditions that are more or less likely to transit into a virtual care modality. Further to this, specific exploratory studies and interventions can then be designed to understand barriers and facilitators to the use of telemedicine in primary and acute care.

I have a few minor comments (see below) to advise the authors in strengthening their manuscript. Wishing you all the best in your research!

Introduction

- (Line 67) Looks like a missing word in "it is also unclear if virtual care *is* able..."

Methods

- Under "Data Sources", how thorough is the coverage of these administrative databases in comparison to Ontario's total healthcare utilisation? It would be good to have an approximation if possible, as that gives the reader an indication of the generalisability of this study's findings.

Discussion

- (Lines 229 - 231) Is this claim necessarily true? I understand that the data sources used in this study may not contain sociodemographic or economic information, but is there a possibility that patients who received the most care before and during the pandemic (including use of virtual care services) simply had more access to care? Does virtual care service use reflect sociodemographic distributions, and is there any data on this in the national or published literature?

- (Line 234) This is a minor point, but it may be useful to use a range of months instead of seasons for the benefit of equatorial or southern hemisphere readers.

- (Lines 236 - 237) What is this government-run platform (I believe this is the first time it has been mentioned in this manuscript)? Is there any reason for the disparity in usage among healthcare professionals? Are there other teleconsult/telehealth platforms not run by the government? A brief elaboration of Ontario's virtual care infrastructure could be useful in strengthening the discussion.

- (Line 246) Minor comment on inconsistent referencing style.

- (Lines 253 - 255) While there is a possibility that this is the case, I would argue whether other health services-related factors could have contributed to this phenomenon. For example, is it possible that many pre-pandemic in-patient visits were actually unnecessary and the pandemic simply led healthcare providers to realise that the same level of care could be provided virtually? This is an important point of further discussion for the topic of healthcare utilisation.

- (Lines 259 - 260) Is there any citable evidence to support this phenomenon in Ontario?

Reviewer #2: Although the undertaken topic is interesting and the use of virtual clinics requires attention from researches, the findings presented in this article add no value to literature and neither to the practice. The paper, in present form lacks novelty and implications. The objective of the paper is also too simplistic.

Additional I shall note, the introduction of the paper lacks presentation of results of other scholars studies relating to the use of virtual clinic during COVID-19.

Reviewer #3: Virtual care use during the COVID-19 pandemic and its impact on healthcare utilization in patients with chronic disease: a population-based repeated cross-sectional study

Below are comments to improve the manuscript;

Re-write the Background: in the abstract

The paper is not well written. Incude the introduction, literature review..

The main objective and research questions to be explored are missing

The conculsion section is missing.

The discussion can be improved.

Several studies on Virtual care use during the COVID-19 pandemic are missing in the manuscript only 10 sources were cited.

Reviewer #4: The authors intended to provide information on the levels of use of virtual care services and of healthcare utilization using the health administrative data. The results showed the increased use of virtual care, and decreased hospitalizations and laboratory testing at the start of the pandemic, while increased later in high virtual care users. This is more a descriptive report. The manuscript was well prepared. Some minor comments were as follows.

1. line 111. patient had a maximum of one virtual care visit were classified into low virtual care users. does this mean it doesn’t count the numbers of in-person visit? what if they have more in-personal care visit? it should be clarified.

2. Figure. it seems that there was clear drop from Jan-March. Any explanation?

3. This manuscript is more a descriptive oriented report. But it would be helpful to provide some public health implications based on the reported results.

6. PLOS authors have the option to publish the peer review history of their article (what does this mean?). If published, this will include your full peer review and any attached files.

Reviewer #1: No

Reviewer #2: No

Reviewer #3: No

Reviewer #4: No

---

## [Author Response · Author response to Decision Letter 0]

20 Feb 2022

Dear Editor and Reviewers,

Thank you for taking the time to review our manuscript. We have taken all your comments into consideration and have made changes to the manuscript. You can see our detailed responses to your comments below. 

We look forward to your decision. 

Best Regards,

Vess Stamenova on behalf of all co-authors

Editorial Requests

Revise the references to include other scholars studies related to the use of virtual clinic during COVID-19.

Please note that we have revised the introduction and the discussion and have included some more recent literature evidence. Tracked changes are included in the document and more detailed responses found below with the reviewers

The work of Dr Bhatia (ref 1) discusses about the potential risks of deferring medical care, but not specifically about ACSC.

Thank you for flagging this issue. We have removed the reference to ACSC.

Line 127: the sum of the four groups of visits is 215 million, instead of 218

Thanks for catching that. We have corrected this.

There is a typo in the numbering of tables (table 2 appears before table 1)

Thanks for noticing this. We have corrected this.

Reviewer 1

(Line 67) Looks like a missing word in "it is also unclear if virtual care *is* able..."

Thank you. We added “was” to the sentence. 

Under "Data Sources", how thorough is the coverage of these administrative databases in comparison to Ontario's total healthcare utilisation? It would be good to have an approximation if possible, as that gives the reader an indication of the generalisability of this study's findings.

We have now added the following statement in the introduction: “In Ontario, the most populated province in Canada, the healthcare system is publicly funded and the data we used covers all healthcare services used(13). It, therefore, presents an opportunity to examine shifts in care at a population-level.”

(Lines 229 - 231) Is this claim necessarily true? I understand that the data sources used in this study may not contain sociodemographic or economic information, but is there a possibility that patients who received the most care before and during the pandemic (including use of virtual care services) simply had more access to care? Does virtual care service use reflect sociodemographic distributions, and is there any data on this in the national or published literature?

That’s true. Low users of the system may be low users due to poor access and if anything the data suggests that the higher users with good access continued to have good access, but those with low use (due to poor access or lack of need) continued to have a low use. We have rephrased our language around that statement now and on p.11 we state:

“This suggests that higher use patients who had access to the healthcare system before the pandemic received similar access to the healthcare system during the pandemic though a combination of care services including virtual care.”

(Line 234) This is a minor point, but it may be useful to use a range of months instead of seasons for the benefit of equatorial or southern hemisphere readers.

Noted. We have changed this now to say May to Sep.

(Lines 236 - 237) What is this government-run platform (I believe this is the first time it has been mentioned in this manuscript)? Is there any reason for the disparity in usage among healthcare professionals? Are there other teleconsult/telehealth platforms not run by the government? A brief elaboration of Ontario's virtual care infrastructure could be useful in strengthening the discussion.

Thanks for noting that. 

Generally, in Ontario access to virtual care before the pandemic was limited to specialists and some primary care physicians. These physicians had access only to a standard provincial videoconferencing platform that they were to use for virtual care visits and they were not allowed to use any other platforms (e.g. Zoom). Once the pandemic started all providers were allowed to use any platform they chose to (by the province creating temporary virtual care billing codes), which allowed much more flexibility for physicians to conduct virtual care. Most physicians used the other platforms and not the government run platform. Psychiatrists who were the highest users of virtual care before the pandemic were already using the government run platform and while most of care in psychiatry was also done through other platforms during the pandemic, some providers likely decided to continue using the government run platform and therefore they were the highest users of that platform. Overall, the platform was used very little, however (less than 6% of all virtual care). 

We have decided to omit that point from the paper, as we don’t think it adds much to our objectives of describing virtual care use or to explaining the data.

(Line 246) Minor comment on inconsistent referencing style.

Thanks for noting that. We have referenced now the correct article in proper referencing style.

(Lines 253 - 255) While there is a possibility that this is the case, I would argue whether other health services-related factors could have contributed to this phenomenon. For example, is it possible that many pre-pandemic in-patient visits were actually unnecessary and the pandemic simply led healthcare providers to realise that the same level of care could be provided virtually? This is an important point of further discussion for the topic of healthcare utilisation.

We agree. We have modified our statements to say that the causes listed here relate more to reductions in total ambulatory care, as opposed to just in-person care and we have added some statements to reflect your suggestions. The paragraph now reads:

“The pandemic led to an initial overall decrease of total ambulatory care across all chronic diseases studied. This reduction in ambulatory visits and testing has been reported elsewhere(4,6), with some reporting more severe impacts in diabetes management(5). It has been suggested that the causes of these reductions in utilization are multifactorial and include patient avoidance of care, increased threshold of hospitalizations from providers, and changes in lifestyle and self-management in the context of lockdown measures and social distancing(6). The decrease in total ambulatory care was a result of a very sharp drop of in-person care due to COVID-19 restrictions. Restrictions on in-person care, however, were accompanied by an increase in virtual care, which ultimately allowed patients with chronic disease to return to the same visit rate as they had before the onset of the pandemic. This was likely due to both patients and providers adopting virtual care as part of their routine care and enabling the conversion of pre-pandemic in-person visits into virtual visits where possible.(19) “ 

- (Lines 259 - 260) Is there any citable evidence to support this phenomenon in Ontario?

Yes. Elective procedures were cancelled during the early stages of the pandemic and there have been reports on changes in physician behavior. We have added this to our manuscript now on p. 13:

“This reduction on non-urgent care was likely partially due to restrictions on elective procedures in the province in effect during the early stages of the pandemic, but it may also be a result of changes in physician prescribing and testing practices(18).”

Reviewer 2

Although the undertaken topic is interesting and the use of virtual clinics requires attention from researches, the findings presented in this article add no value to literature and neither to the practice. The paper, in present form lacks novelty and implications. The objective of the paper is also too simplistic.

Additional I shall note, the introduction of the paper lacks presentation of results of other scholars studies relating to the use of virtual clinic during COVID-19.

Please note that we have revised the introduction and the discussion and have included some more recent literature evidence. Tracked changes are included in the document. 

We have added another four references supporting the reduction in preventative measures in Canada and the initial reduction in hospitalizations seen early in the pandemic.

Reviewer 3

Re-write the Background: in the abstract

We have clarified that we are reporting a population-level study that encompasses data from most of Ontario’s population.

The paper is not well written. Incude the introduction, literature review..

We have revised the introduction extensively and have included some more recent references to it.

The main objective and research questions to be explored are missing

The objective of the study is listed in the last paragraph of the introduction section:

“The goals of our study were to provide a description of the levels of use of virtual care services, relative to in-person care in patients with chronic disease across Ontario, Canada and to describe levels of healthcare utilization in low versus high ¬virtual care users.”

The conculsion section is missing.

The conclusion statement was the last paragraph. This has now been formatted to PLoS One style and is separated in a distinct section with its own subheading.

The discussion can be improved.

We have made changes to the discussion based on the feedback provided by the other reviewers.

- We have made changes to comments regarding reductions in total (as opposed to in-person) ambulatory care

- We have added separate explanation regarding reductions in in-person care alone, incorporating one of the reviewer’s comments about virtual care being adopted more by both physicians and patients

- We have added supporting literature on the idea that patients that received more virtual care had poorer health overall.

- We have added comments around the greater public health implications of our findings.

Several studies on Virtual care use during the COVID-19 pandemic are missing in the manuscript only 10 sources were cited.

Please note that we have revised the introduction and the discussion and have included 8 more references. Tracked changes are included in the document.

Reviewer 4

line 111. patient had a maximum of one virtual care visit were classified into low virtual care users. does this mean it doesn’t count the numbers of in-person visit? what if they have more in-personal care visit? it should be clarified.

Please note that we have clarified this on p.6, stating:

“To examine differences in healthcare utilization between patients who use virtual care and those who do not, we classified patients into two groups based on their use of virtual care. Low virtual care users had at least one visit (virtual or in-person) after the onset of the pandemic (March 14, 2020) and they could have a maximum of one virtual care visit during the entire period. Patients who had no virtual care visits were also included in the low virtual care group, but they had to have at least one in-person visit in order to be included. This means we excluded patients who did not receive care at all during the pandemic period (after March 14, 2020). Patients in the high virtual care use group had to have at least 2 virtual care visits. There was no limit on the number of in-person visits patients could have in either group.”

Figure. it seems that there was clear drop from Jan-March. Any explanation?

The drops that seem to be in Jan-March are actually in late December and are due to holiday closures. 

This manuscript is more a descriptive oriented report. But it would be helpful to provide some public health implications based on the reported results. 

Please note that we have included the following in our conclusion statement:

“These findings suggest that proper frequency of visits of chronic disease patients can be maintained through a mix of in-person and virtual visits even in cases where the disease severity is higher. Long-term studies should examine, however, whether the quality of care received through a mixed in-person and virtual care model is the same as that received through in-person care alone in order to inform policy decisions about their continued use in the healthcare system.”

---

## [Editor Report · Decision Letter 1]

5 Apr 2022

Virtual care use during the COVID-19 pandemic and its impact on healthcare utilization in patients with chronic disease: a population-based repeated cross-sectional study.

PONE-D-21-24910R1

Dear Dr. Stamenova,

We’re pleased to inform you that your manuscript has been judged scientifically suitable for publication and will be formally accepted for publication once it meets all outstanding technical requirements.

However,  there are some very minor flaws to be corrected

I have noticed several typos. The first one is relevant. Please revise the whole manuscript:

Line 101: the figures of the percentage are missing (“databases cover ??% of healthcare”)

Line 71: “a” after the full stop

Line 168: This title should maintain a similar format to the other ones.

Line 171: There is an unnecessary slash (“/angina”)

Line 255: There is a repeated full stop.

Aditionally, the results for hospitalizations might produce some confusion (page 9). In lines 189; 193; 194; 196; and 197 the figures correspond to volumes of hospitalizations. It would be clearer if were referred as “admissions” instead of “visits”.

Kind regards,

Juan F. Orueta, MD, PhD

Academic Editor

PLOS ONE

---

## [Editor Report · Acceptance letter]

14 Apr 2022

PONE-D-21-24910R1 

Virtual care use during the COVID-19 pandemic and its impact on healthcare utilization in patients with chronic disease: a population-based repeated cross-sectional study. 

Dear Dr. Stamenova:

I'm pleased to inform you that your manuscript has been deemed suitable for publication in PLOS ONE. Congratulations! Your manuscript is now with our production department. 

Kind regards, 

on behalf of

Dr. Juan F. Orueta 

Academic Editor

PLOS ONE